# The magnitude of hypertension and associated factors among clients on highly active antiretroviral treatment in Southern Ethiopia, 2020: A hospital-based cross-sectional study

**Kaleegziabher Lukas**[1◉]**, Endrias Markos**[2◉]**, Fanuel Belayneh**[2◉]**, Akili Habte**[1]*

1 School of Public Health, College of Medicine and Health Sciences, Wachemo University, Hosanna, Ethiopia, 2 School of Public Health, College of Medicine and Health Sciences Hawassa, Hawassa University, Hawassa, Ethiopia

◉ These authors contributed equally to this work.
* akliluhabte57@gmail.com

**Data Availability Statement:** All relevant data are within the manuscript and its Supporting Information files.

## Abstract

### Introduction

Following the introduction of Highly Active Anti Retro Viral Treatment (HAART), the survival of people living with HIV/AIDS (PLHIV) has improved. However, hypertension remains a major challenge for people living with HIV. Very little effort has been made to examine the magnitude of hypertension and its contributing factors among clients receiving HAART, particularly in southern Ethiopia. Hence, the current study aimed at determining the frequency of Hypertension and associated factors among clients receiving HAART at Wachemo University Nigist Eleni Mohammed Memorial Referral Hospital, southern Ethiopia, 2020.

### Methods

A hospital-based cross-sectional study took place from January 20- March 20, 2020. A systematic sampling technique was employed in the selection of 397 clients. Interviewer administered pretested structured questionnaire was used for data collection. Blood pressure and anthropometric parameters of PLHIV were measured. The data was encoded and entered using Epi Data Version 3.1 and exported to SPSS version 23 for analysis. Then bivariable and multivariable logistic regression analyses were used to identify associated factors. Adjusted Odds Ratio (AOR) with 95% CI was used to present the estimated effect size and declare the presence of statistically significant association respectively.

### Results

The magnitude of hypertension among clients on HAART was 11.0% 95% CI [7.93, 14.04]. Being on HAART for at least 60 months (AOR: 2.57, 95% CI: 1.24–5.21), being on TDF/3TC/EFV combination (AOR: 4.61, 95% CI: 2.52–8.3), and high alcohol consumption (AOR:

**Funding:** The author(s) received no specific funding for this work.

**Competing interests:** The authors have declared that no competing interests exist.

**Abbreviations:** AIDS, Acquired Immuno Deficiency Syndrome; AOR, Adjusted Odds Ratio; ATV, Atazanavir; AZT, Zidovudine; BMI, Body Mass Index; D4T, Stavudine; EFV, Efavirenz; HAART, Highly Active Anti Retro Viral Treatment; NEMMRH, Nigist Elleni Mohammed Memorial Referral Hospital; NVP, Nevirapine; PLWHIV, People Living with HIV; SPSS, Statistical product and service solutions; WHO, World Health Organization.

4.31, 95% CI: 1.84–10.02) were identified as significant predictors of hypertension among clients on HAART.

## Conclusion and recommendation

The magnitude of hypertension in the study area was in a considerable state to plan and implement intervention measures. For those clients who have received TDF/3TC/EFV and TDF/3TC/NVP and those who have been on HAART for 60 months, a strong emphasis should be placed on planning a strict follow-up. A concerted effort among health care providers is needed through counseling and education to discourage the habit of high alcohol consumption among clients.

## Introduction

The Human Immunodeficiency Virus (HIV), a very common communicable disease, remains one of the major public health challenges [1]. By the end of 2018, about 37.9 million people worldwide were living with HIV, with sub-Saharan Africa being the most affected region, with 25.6 million people living with HIV in the region, sharing two-thirds of all cases worldwide [2]. In Ethiopia, 722,248 people were living with HIV, of which approximately 23,000 were newly infected by annual AIDS-related deaths of 11,000, and prevalence and incidence were 0.4% and 1% respectively [3]. Acquired Immune Deficiency Syndrome (AIDS) has become central to many global concerns and has reached epidemic proportions in some parts of the world [4].

Hypertension is an elevated systemic blood pressure condition that is typically asymptomatic. It is a major cardiovascular risk factor that is closely associated with severe complications such as coronary artery disease, stroke, heart failure, and renal failure [5]. HIV can cause inflammation in the blood vessels, boost atherosclerosis, and the formation of high-risk plaques, which increases the risk of cardiovascular diseases [6]. Besides, some antiretroviral medications used to treat HIV also increased cholesterol levels, abdominal fat, and blood pressure [7].

While, since the beginning of Highly Active Antiretroviral Therapy(HAART) has made great progress in improving the longevity and quality of life of people living with HIV (PLWHIV), treatment is not risk-free or has side effects. Complications such as dyslipidemia and related high blood pressure are the most commonly encountered complications of ART, and it has a well-established increased risk of cardiovascular morbidity in people infected with HIV [8–11]. In people living with HIV who receive ART, death results from cardiovascular complications such as hypertension related to the virus, host, and ART factors [9–11].

Hypertension in HIV-infected adults is linked to a higher incidence of persistent proteinuria, coronary heart disease, and myocardial infarction than in subjects not infected with HIV [12]. Of all the comorbidities in people living with HIV, hypertension appeared among the main causes of non-AIDS related mortality [13]. The prevalence of hypertension among people living with HIV is 4.7–54.4% in high-income countries and 8.7–45.9% in low- and middle-income countries [14].

Studies in people infected with HIV have reported various metabolic changes associated with HAART, including alterations in lipid and glucose metabolism, peripheral arterial disease, and coronary artery disease. Moreover, the simultaneous occurrence of hypertension and

HIV among people infected with HIV can complicate the management of HIV infection, increasing the risk of morbidity and mortality of these people [15, 16].

Although the burden of non-communicable diseases, especially hypertension, is significant in people living with HIV/AIDS, few studies have shown that ART has had a negative effect. To date, very limited attempts have been made in Ethiopia to study the magnitude of hypertension and contributing factors in ART clients [11, 17]. As a result, less focus was put on solutions to avoid hypertension among those clients. To date, the diagnosis, prevention, and management of hypertension are not routine tasks and have not focused on risk reduction among clients receiving ART.

Also, certain clinical factors such as Lypodystrophy and individual factors such as knowledge and attitude regarding the prevention of hypertension were not addressed in existing studies [11, 17, 18]. Furthermore, the effect of other variables, such as the type of current ART regimen, the change in regimen, and adherence to ART on hypertension, was not well addressed in prior studies. Hence, this study aimed at providing a clear picture of the magnitude of hypertension and its determinants among clients on HAART in Nigist Eleni Mohammed Memorial Referral Hospital, southern Ethiopia.

## Material and methods

### Study design and setting

A hospital-based cross-sectional study was conducted at Wachemo University Nigist Elleni Mohammed Memorial Referral Hospital (NEMMRH) from January 20 to March 20, 2020. The hospital is located 232 km from Addis Ababa, the capital of Ethiopia, and 157 km from Hawassa, the capital of the southern region. NEMMH under Wachemo University has 201 beds giving services in the 4 major departments namely: Surgical, Gynecologic and Obstetric, Internal Medicine, and Pediatrics. According to information obtained from the hospital's Health Management Information System office, the hospital began delivering the HAART service in 2006 GC. It provides services with 4 general practitioners, 3 health officers, 3 nurses, and 3 pharmacists who have been trained in ART. Now it is among the largest ART centers in southern Ethiopia serving over 1,200 adult clients receiving HAART.

### Populations of the study

The source populations were all enrolled adult clients on HAART in the hospital. The study population was the selected clients aged 18 and over who met all inclusion criteria during the data collection period. Adult clients on HAART were included in the study. Women who had a confirmed pregnancy and had been on hormonal contraception for more than six months, as well as seriously ill patients, were excluded from the study.

### Sample size determination

The sample size for the first objective was determined by using the single population proportion formula. By using the following parameters: the estimated proportion of hypertension among clients receiving HAART in Eastern Ethiopia was 12.7% [11], the margin of error 3% since the prevalence is small, 95% level of confidence, and 10% non-response rate, the ultimate sample size for the study were 519. Correction formula is applied since the population is less than 10,000 and hence the final sample size for the first objective was 352. Secondly, a double population formula was employed by using the Stat Calc menu of Epi Info version 7 with consideration of factors associated with hypertension among clients on ART. Among those factors selected, the largest sample size(n = 620) was obtained by considering percent of hypertension

in unexposed(i.e. current CD4 count <500cells/mm3 = 90%), AOR of 2.7 [11]. Besides other parameters like; 80% power, 95% confidence level, 5% degree of precision, the ratio of unexposed to expose equally to 1, non-response rate of 10%. Again correction formula is applied and the final sample size for the second objective was 397. Finally, the sample size obtained using two population proportion considerations (n = 397) was larger than the sample size for a single population proportion (n = 352) and it was taken as the final sample size for the study.

## Sampling procedures

The average number of clients visiting the HAART clinic within the past six months was computed and it was 628. To include study participants, a systematic sampling technique was employed. The interval (K) was determined by dividing the average number of clients visiting the clinic per month by the sample size (i.e. 628/397≈2). Finally, after randomly selecting the first client, every other 2nd adult client was included and interviewed.

## Data collection tools, methods, and personnel

Structured questionnaires were developed after an in-depth review of the literature on the area of interest [11, 16, 19, 20]. Blood pressure was measured based on the new WHO recommendation [19], where, the participant was in a sitting position and his/her back being supported, leg uncrossed, empty bladder, not talking, and the patient has taken an adequate rest for 5 minutes and was measured by two BSc Nurses using a mercury sphygmomanometer and stethoscope having adult size cuff. The BP measurement was consistently taken 2 times from the left arm 5 minutes apart and the second reading was taken to be estimated and used in the analysis.

All anthropometric measurements were taken twice and the average was taken. The weight of the participants was taken using the standard beam balance and the scale was verified and calibrated at zero before each measurement. Participants' weight was measured after removing heavy clothing and recorded to the nearest 0.1Kg. Height measurement of participants was taken using the standard measuring scale. Participants' take off their shoes, stand erect, in a horizontal plane. The occiput, shoulder, buttocks, and heels touched the measuring board, and height was recorded to the nearest 0.1cm. Waist circumference was measured at the midpoint between the lowest margin of the last palpable rib and the top iliac crust while the patient is breathing and standing and was categorized according to the WHO standards. Hip circumference was measured at the point yielding the maximum circumference over the buttocks with the tape in a horizontal plane, touching but not compressing the skin. Waist to Hip Ratio (WHR) was calculated as mean waist circumference divided by mean hip circumference and to interpret the result WHO cutoff was used for males and females [11].

In addition, the questionnaire comprised of multiple segments: socio-demographic, behavioral, and clinical characteristics of the respondents. WHO standardized tool was used to assess behavioral and lifestyle factors. Stress was assessed using Cohen's 10 items Perceived Stress Scale (PSS) [21]. Diet-related questions were assessed using a Guideline, Dietary Approach to Stop Hypertension (DASH). Secondary data like WHO staging, baseline CD4 cell count, baseline height and weight, baseline viral load, Level of HDL, and level of LDL were extracted from client medical records. Data were collected by a face-to-face interview on exit, by two trained BSc nurses with the supervision of two public health officers.

## Data quality management

The questionnaire used to collect the data was first prepared in English and translated into the local language by an expert in that language, then translated into English to ensure consistency

with the original meanings. A pre-test was carried out in 5% of the sample size (20 clients) at the Worabe Comprehensive Specialized Hospital. A one-day training focused on: the objective of the study, data collection methods, and ethical issues was provided to data collectors and supervisors. On the date of data collection, the data were checked for completeness and consistency. The data collected has been checked for completeness and consistency on the date of data collection. The principal investigator and supervisor supervised the overall data collection process on daily basis. To mitigate a biased estimation weight scale was calibrated to zero before each measurement and tested with a standard scale.

## Variables of the study

Hypertension was classified as a BP measure of Systolic $\geq$ 140 mmHg and /or/ Diastolic $\geq$ 90 mmHg [19].

Cigarette Smoker: A study participant is considered a cigarette smoker if he/she smokes now or has smoked cigarettes in the past, daily or less than daily [22].Good physical activity: A study participants were considered to be having good physical activity if they had at least 150 minutes of moderate physical activity or at least 75 minutes of vigorous physical activity during work, Transportation, and leisure all week long [23].

Good knowledge about prevention of hypertension: A study participant was considered to have a good knowledge of hypertension prevention if he or she scored above the average of the 10 knowledge questions [24].

Alcohol drinking: considered as no/mild, moderate, and high alcohol consumer if he/she scored < 8, 9–14 and scored more than 15 of the questions on alcohol consumption respectively [25].

Stress level: A study participants should be considered as low, medium, and high-stress levels if they scored 0–13, 14–26, and 27–40 out of 40 Perceived Stress Scale (PSES) questions, respectively [21].

Khat Chewer: One study participant who chews khat even once within 30 days before the study was considered a khat chewer.

Lipodystrophy: A study participant with fat loss in the face or arms and/or fat gain in the back or at the base of the neck, increased breast size, decreased fat on the buttocks or legs will be considered Lipodystrophy.

A negative attitude about prevention of hypertension: if he/she scored lower than the average of the 8 attitude questions. [24].

Good adherence to HAARTs/good treatment interaction-: if he /she didn't miss even a single dose of the prescribed regimens in the past one week [26].

Regimen Change: A study participant was considered as having a history of regimen change if he/she switched or substitute for at least one drug from the original HAART regimen for the first time due to several reasons.

## Data analysis

After verifying its completeness and consistency, the data was entered using EpiData version 3.1 and exported to SPSS version 23 for analysis. Descriptive statistics such as frequency distributions, mean, median standard deviation, and inter-quartile range have been calculated to quantify the variables. Bivariable and multivariable logistic regression was used to identify factors associated with hypertension. In the bivariable analysis, each explanatory variable was tested for the presence of an association with the outcome variable and the significant variables at P <0.25 were candidates for multivariable logistic regression. Lastly, independent variables with P <0.05 in multivariable logistic regression were reported as statistically significant

variables with the dependent variable. The adjusted odds ratio (AOR) and with it 95% CI were used to report the association.

### Ethical approval and consent to participate

Ethical clearance was obtained from the Institutional Review Board of Hawassa University College of Medicine and Health Sciences and a support letter was written to NEMMRH officials and concerned bodies about the purpose and importance of the study. Written informed consent for the research was given by all participants. Respondents were also well informed that the information provided during the study would only be used for research purposes and would not be disclosed to anyone other than the research team. To keep the questionnaire anonymous, a unique identification number was issued. Confidentiality and privacy were preserved throughout all the study procedures. Also, all hypertensive and hypotensive patients identified during BP measurement were linked to the relevant care provider to receive adequate treatment.

## Results

### Socio-demographic characteristics of respondents

Of the total of 397 study participants sampled, 382 participated in the study, which yielded a response rate of 96.2%. The age of respondents ranged from 19–63, with a median age of 35 (IQR: 29–43), with the majority (60.8%) belonging to the 31–50 age group. Over half (53.9%) Of the respondents were women. Concerning educational attainment 138 (36.1%) had completed high school. Also, 245 (64.1%) were Protestant by religion and 259 (67.8%) were from the Hadiya ethnic group (Table 1).

### Clinical profiles of respondents

Close to nine in ten respondents (88.2%) had a reference CD4$^+$ cell count of $<$200 cells/mm3 and 357 (93.5%) had a recent CD4$^+$ cell count of $\geq$ 200 cells/mm3 or higher to date. A recent viral load of 50–1000 copies/ml was recorded for 335 (87.7%) of participants. Lipodystrophy was recorded for 86 clients (22.5%). Two hundred fifty-two (66.5%) respondents had a history of regimen change. Whereas, 120(30.2%) were currently on AZT/3TC/EFV. About three-quarters (74.1%) of respondents reported that they sustained treatment interaction during the preceding 7 days. Two hundred ninety-five (92.7%) had an HDL level of $\geq$ 40mg/dl, with a mean of 63(SD± 10) mg/dl. A high LDL level of $\geq$130mg/dl was recorded among 15(4.8%) clients. Also, 300 (78.5%) participants reported that their BP had already been measured monthly. Document review showed that 164(42.9%) of respondents had at least one type of opportunistic infection, among whom 101(61.5%) had skin disorders (Table 2).

### Anthropometric measurement

The mean (± SD) for the baseline weight of participants was 42.7 (± 6.3) Kg, whereas, the current weight being 56 (± 4.2) Kg. The mean (±SD) height was 1.6 (± 0.11) M. Three hundred twelve (81.7%) were underweight during initiation of HAART with BMI$<$18.5Kg/M$^2$. Thirteen male respondents (3.4%) had a high Waist-to-Hip ratio (WHR) of $\geq$0.9% (Table 3).

### Behavioral characteristics of respondents

A good level of physical activity was reported by 200(52.4%) of respondents. Three hundred sixty-one (94.5%) were nonsmokers and for the item asking about alcohol consumption frequency, 266 (69.6%) replied that they had a habit of low-level alcohol drinking in the past year. More than seven in ten (71.7%) had no prior history of chewing khat and 228 (59.7%) were

**Table 1. Socio-demographic characteristics of clients receiving HAART at Nigist Elleni Mohammed Memorial Referral Hospital (NEMMRH), Southern Ethiopia, January 20 –March 20, 2020.**

| Variable Categories(n = 382) | Frequency | Percentage |
|---|---|---|
| **Age in years** | | |
| 19–30 | 114 | 29.8 |
| 31–50 | 232 | 60.8 |
| > = 51 | 36 | 9.4 |
| **Sex** | | |
| Male | 176 | 46.1 |
| Female | 206 | 53.9 |
| **Religion** | | |
| Protestant | 245 | 64.1 |
| Orthodox | 66 | 17.3 |
| Muslim | 65 | 16.9 |
| Catholic | 6 | 1.5 |
| **Ethnicity** | | |
| Hadiya | 259 | 67.8 |
| Kembata | 89 | 23.2 |
| Amhara | 34 | 8.9 |
| **Educational Status** | | |
| No formal education | 80 | 20.8 |
| Primary education | 129 | 33.7 |
| Secondary education | 138 | 36.1 |
| College and above | 36 | 9.4 |
| **Marital Status** | | |
| Married | 157 | 41.1 |
| Single | 120 | 31.4 |
| Divorced | 47 | 12.3 |
| Widowed | 58 | 15.2 |
| **Occupation** | | |
| Housewife | 119 | 31.2 |
| Farmer | 58 | 15.2 |
| Merchant | 55 | 14.4 |
| Daily laborer | 82 | 21.4 |
| Civil Servant | 68 | 17.8 |
| **Residence** | | |
| Urban | 250 | 65.4 |
| Rural | 132 | 34.6 |
| **Average Monthly Income** | | |
| < = 500ETB | 164 | 43 |
| 501-1500ETB | 146 | 38.2 |
| > = 1501 ETB | 72 | 18.8 |

with low-stress levels. Regarding the dietary practice, 235(59.3%) reported that they had a vegetable intake of 1–7 times per week (Table 4).

## Knowledge and attitude towards hypertension prevention measures

To assess their level of knowledge about hypertension prevention, the study participants were asked questions and 236 (61.8%) had good knowledge about hypertension prevention. Besides, 128 (33.5%) had a negative attitude towards hypertension prevention (Fig 1).

**Table 2. Clinical profiles of clients on HAART at NEMMRH, Southern Ethiopia, January 20-March 20, 2020.**

| Clinical characteristics of respondents | Frequency | Percent |
|---|---|---|
| Baseline $CD_4^+$ cell count(n = 382) | | |
| < 200 cells/ $mm^3$ | 337 | 88.2 |
| ≥ 200 cells/ $mm^3$ | 45 | 11.8 |
| Current $CD_4^+$ cell Count(n = 382) | | |
| <500 cells/ $mm^3$ | 25 | 6.5 |
| ≥500 cells/ $mm^3$ | 357 | 93.5 |
| Duration of HAART(n = 382) | | |
| ≥60 months | 137 | 35.9 |
| < 60 months | 245 | 64.1 |
| Lipodystrophy(n = 382) | | |
| Yes | 86 | 22.5 |
| No | 296 | 77.5 |
| History of Opportunistic Infection(n = 382) | | |
| No | 218 | 57.1 |
| Yes | 164 | 42.9 |
| Family History of NCDs(n = 382) | | |
| Yes | 54 | 14.1 |
| Don't know | 52 | 13.6 |
| No | 276 | 72.3 |
| Regimen Change(n = 382) | | |
| Yes | 252 | 65.9 |
| No | 130 | 34.1 |
| Current HAART Regimen(n = 382) | | |
| TDF/3TC/EFV | 68 | 17.8 |
| TDF/3TC/NVP | 94 | 24.6 |
| AZT/3TC/EFV | 120 | 31.4 |
| AZT/3TC/NVP | 100 | 26.2 |
| Adherence to HAART(n = 382) | | |
| Yes | 283 | 74.1 |
| No | 99 | 25.9 |
| Current Viral Load(n = 382) | | |
| 50–1000 copies/ml | 335 | 87.7 |
| 1001-10000copies/ml | 47 | 12.3 |
| HDL (n = 318) | | |
| <40mg/dl | 23 | 7.3 |
| > = 40 mg/dl | 295 | 92.7 |
| LDL (n = 318) | | |
| > = 130mg/dl | 15 | 4.8 |
| <130 mg/dl | 303 | 95.2 |

## The magnitude of hypertension among clients on HAART

Of a total of 382 respondents on HAART, 42 (11.0%) were diagnosed with hypertension. The mean systolic and diastolic blood pressure of participants was 108 mmHg (SD±15) and 78mmHg (SD±10) respectively. Thirty (7.8%) were having a systolic BP measure of ≥140mmHg and 12(3.2%) were having a diastolic BP measure of ≥90mmHg (Fig 2).

**Table 3. Anthropometric related characteristics of respondents receiving HAART at NEMMRH, Southern Ethiopia, and January 20-March 20, 2020.**

| Anthropometric variables (n = 382) | Frequency | Percentage |
|---|---|---|
| **Baseline BMI** | | |
| <18.5 Kg/M$^2$ | 312 | 81.7 |
| 18.5–24.9 Kg/M$^2$ | 65 | 17.0 |
| ≥25Kg/M$^2$ | 5 | 1.3 |
| **Current BMI** | | |
| <18.5Kg/M$^2$ | 65 | 17.0 |
| 18.5–24.9Kg/M$^2$ | 313 | 82.0 |
| ≥25Kg/M$^2$ | 4 | 1.0 |
| **Male Waist to Hip Ratio** | | |
| ≥ 0.9 | 13 | 7.3 |
| < 0.9 | 163 | 92.7 |
| **Female Waist to Hip Ratio** | | |
| ≥ 0.85 | 3 | 1.4 |
| < 0.85 | 203 | 98.6 |

## Factors associated with hypertension among clients on HAART

A multivariable logistic regression analysis was carried out and four variables namely; duration of staying on HAART, baseline BMI, high alcohol consumption, and the current HAART regimen were significantly associated with being hypertensive. This study has shown that staying on HAART for a long time increases the likelihood of developing hypertension. Clients who had been taking HAART for ≥60 months were 2.57 times more likely to develop hypertension than their counterparts (AOR: 2.57, 95% CI: 1.24–5.21). Besides, those who consumed high levels of alcohol were 4.3 times more likely to develop hypertension than those who consumed low levels (AOR: 4.31, 95% CI: 1.84–10.02). It was also determined that the type of current regimen taken by clients showed a significant association with hypertension. Those, who were on TDF/3TC/EFV combinations had 4.6 times increased odds of developing hypertension when compared to those on AZT/3TC/NVP (AOR: 4.61, 95% CI: 2.52–8.3) (Table 5)

## Discussion

This transversal hospital study was attempted to assess the extent of hypertension and its associated factors in HAART clients. The magnitude of hypertension in the study setup was 11.0%.

**Table 4. Diet related characteristics of clients receiving HAART at NEMMRH, Southern Ethiopia, January 20-March 20, 2020.**

| Variable categories (n = 382) | Frequency | Percent |
|---|---|---|
| **Fruit Intake** | | |
| 1–7 times/week | 183 | 47.9 |
| None | 199 | 52.1 |
| **Vegetable Intake per week** | | |
| 1–7 times | 234 | 61.3 |
| None | 148 | 38.7 |
| **Type of Oil used to prepare a meal** | | |
| Saturated | 230 | 60.2 |
| Unsaturated | 152 | 39.8 |

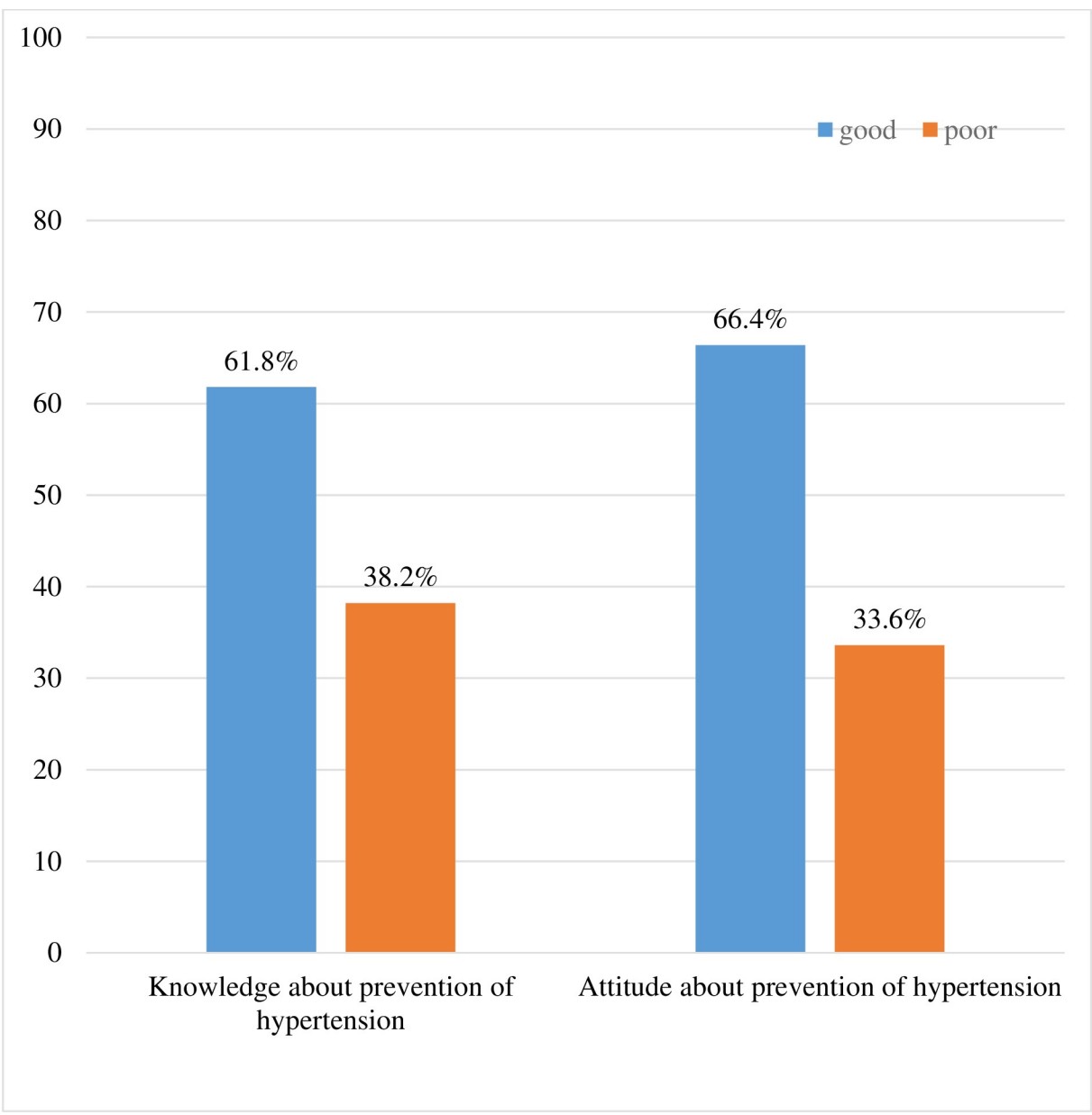

**Fig 1. Knowledge and attitude of clients receiving HAART towards prevention of hypertension, Southern Ethiopia, January 20-March 20, 2020.**

The factors identified significantly associated with hypertension were, being on TDF/3TC/ EFV and TDF/3TC/NVP combinations, high alcohol consumption, and being on HAART for 60 months and more. the magnitude of hypertension in this study was consistent with the findings from Kenya and Eastern Ethiopia in which 13.4% and 12.7% of clients receiving HAART were hypertensive [11, 27]. While the magnitude is lower than studies carried out in the USA (31.0%), Nigeria (40.9%), Uganda (24.8%), southern Ethiopia (15.9%), and southwestern Ethiopia (16.0%) [16–18, 28, 29]. The difference can be attributed to the age differences of participants in which 61% of respondents in this study were aged 31–50 years, while the age group was higher(> = 50yrs) in those studies with higher magnitude, this, in turn, raises the risk of

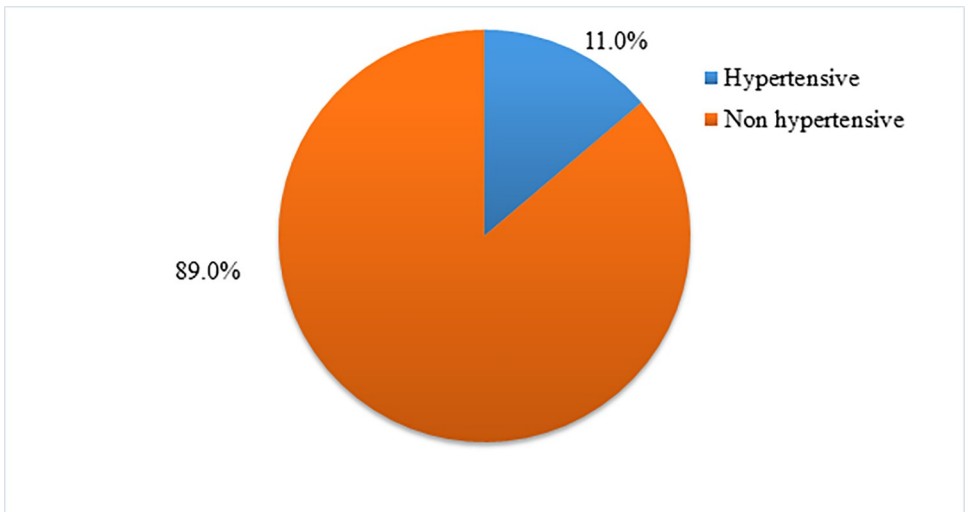

**Fig 2. The magnitude of hypertension among clients on HAART at NEMMRH from January 20-March 20, 2020.**

hypertension [17, 18, 28, 29]. In addition, the hyperlipidemia status in this study was 12%, which is lower than previous studies done elsewhere [16, 29], which can lead to increased levels of lipids in blood vessels, which in turn increases the risk of hypertension [4, 30]. Furthermore, the type of HAART taken by clients may be the potential cause of the variance. The majority of respondents to the current study are taking HAART which are nucleoside reverse transcriptase (NRTI) inhibitors, while clients of other studies with high magnitude were taking protease inhibitors (IP), these drugs are known to raise lipid levels and thus cause hypertension [29, 31].

This study found that taking HAART over a long period increases the likelihood of developing hypertension. Clients who had been on HAART for ≥60 months were more likely to develop hypertension than their counterparts. This is consistent with studies carried out elsewhere in Ethiopia [11, 18]. This can be explained by the fact that clients taking HAART for a long time may develop abdominal fat and cholesterol, which can facilitate the onset of hypertension by changing the normal physiological conditions of the body [7]. Besides, some HAART facilitates the development of high blood pressure, due to the mechanism of change in lipid and glucose metabolism resulting in an increase in peripheral and coronary heart disease over time [4, 15]. This finding contradicted a study in southern Ethiopia that found no link between the duration of ART and hypertension [17]. This discrepancy could be due to the threshold differences used for the duration of HAART, in the current study it was ≥60 months, but in the previous study, it was ≥ 4 years. Furthermore, the discrepancy can be explained by differences in physical activity and work intensity. In the current study, the proportion of participants with good physical activity was 52.4%, versus 69% in the previous one. As a result, it is essential to implement intervention measures and focus on clients who are taking HAART to reduce the occurrence of hypertension.

The present study has shown that high alcohol intake is associated with hypertension. Those who drank a lot of alcohol were 4.3 times more likely to develop hypertension than those who drank little. This finding was supported by a similar study done in Eastern Ethiopia [11]. Alcohol increases BP, through its adverse effect on blood vessels that paves the way for hypertension. The results of this study contradict studies in Uganda and Tanzania which found no link between alcohol use and hypertension [28, 32]. This discrepancy can be explained by the type of tool used to determine the level of alcohol consumed, this study used

**Table 5. Bivariable and multivariable logistic regression analysis of factors associated with hypertension among clients receiving HAART at NEMMRH from January 20-March 20, 2020.**

| Respondents' characteristics | Hypertension | | COR(95%CI) | AOR(95%CI) |
|---|---|---|---|---|
| | Yes (%) | No (%) | | |
| **Age group** | | | | |
| 18–30 | 2(5.7) | 33(94.3) | 1 | 1 |
| 31–50 | 13(5.9) | 205(94.1) | 1.046(0.1,2.19) | 0.423(0.08,2.06) |
| > = 51 | 27(20.9) | 102(79.1) | 4.36(0.5,2.06) | 0.742(0.33,1.62) |
| **Sex** | | | | |
| Female | 26(12.7) | 179(87.3) | 1.45(0.75,2.80) | 1.54(0.741,3.20) |
| Male | 16(9.1) | 161(90.9) | 1 | 1 |
| **Baseline BMI** | | | | |
| <18.5 Kg/ M$^2$ | 6(9.7) | 56(90.3) | 1 | 1 |
| > = 18.5 Kg/ M$^2$ | 36(11.2) | 284(88.7) | 2.64(0.90,4.80) | 1.75(0.75,3.35) |
| **Duration of HAART** | | | | |
| > = 60 months | 22(16.1) | 115(83.9) | 2.15(1.0,4.01)* | 2.57(1.24,5.2) ** |
| < 60 months | 20(8.2) | 225(91.8) | 1 | 1 |
| **Regimen Change** | | | | |
| Yes | 32(12.7) | 220(87.3) | 1.70(0.80,3.58)* | 1.27(0.55,2.89) |
| No | 10(7.7) | 120(92.3) | 1 | |
| **Current HAART Regimen** | | | | |
| TDF/3TC/EFV | 14(20.6) | 54(79.4) | 3.44(1.30,9.06)* | 4.61(2.52,8.3)** |
| TDF/3TC/NVP | 15(15.9) | 79(84.1) | 2.52(0.98,6.49)* | 2.36(1.7,5.8)** |
| AZT/3TC/EFV | 6(5.0) | 114(95.0) | 0.69(0.22,1.2) | 1.4(0.40,4.79) |
| AZT/3TC/NVP | 7(7.0) | 93(93.0) | 1 | 1 |
| **Alcohol Drinking** | | | | |
| High | 19(16.4) | 97(83.6) | 2.06(1.07,3.97)* | 4.3(1.84,10.2)** |
| Low | 23(8.7) | 243(91.3) | 1 | 1 |
| **Waist to Hip Ratio** | | | | |
| > = 0.85 | 2(28.6) | 5(71.4) | 3.33(0.62, 17.7)* | 0.38(0.05,2.81) |
| <0.85 | 40(10.7) | 335(89.3) | 1 | 1 |
| **Fruit Intake** | | | | |
| 1-7/week | 27(13.6) | 172(86.4) | 1.75(0.90,3.42)* | 1.62(0.74,3.53) |
| None | 15(8.2) | 168(91.8) | 1 | 1 |

* Those Variables statistically significant at p-value <0.25 in bivariable analyses.

** Those Variables statistically significant at p-value <0.05 in multivariable analyses.

the WHO tool for alcohol consumption and disorders. The level of knowledge of the participants may also account for the difference. Under the present study, a significant number of participants had poor knowledge about preventing hypertension which could lead to them not being able to be restrained and protect themselves from what is harmful to their health and this will expose them to hypertension. Therefore, a concerted effort among health care providers is needed through counseling and education to discourage the habit of high alcohol consumption among clients.

In this study, the type of HAART regimen currently used increased the risk of hypertension. In particular, respondents who took the TDF/3TC/EFV and TDF/3TC/NVP combination were 4.6 and 2.3 times more likely to develop hypertension than those clients with AZT/3TC/NVP. This was supported by a similar study conducted in Cameroon, in which the

regimens, TDF/3TC/EFV and TDF/3TC/NVP were positively associated with hypertension [20]. This may be due to the nature of protease inhibitor(IP) regimens which are known to increase lipid levels and therefore cause hypertension [29, 31]. Therefore, it is essential to plan strategies to reduce hypertension in clients who receive those selected HAART regimens.

Unlike many studies in Senegal, Malawi, Tanzania, and southern Ethiopia [11, 32–34], baseline BMI is not associated with hypertension. This may be because, in this study, about 82% of study participants had a current BMI of $<18.5 Kg/m^2$. Moreover, most of the study participants were underweight when they began HAART in this study. In contrast, a higher proportion of participants were obese in previous studies, 61.1% and 25%, respectively, in Tanzania and Malawi [32, 34], and just 2.5% in the current study. Another possible rationale for the lack of association in the current study could be a lower state of hyperlipidemia (4.7%) compared to a similar study in Tanzania (61.3%) [32] since Dyslipidemia facilitates the development of atherosclerosis and plaque formation in blood vessels predisposing to hypertension [4, 8].

Unlike other studies, the current study has sought to address factors that have not been addressed in other studies such as recent viral load, opportunistic infections, adherence with HAART, and knowledge and attitudes on preventing hypertension. Although considerable efforts have been made to minimize the bias effect in this study, it will not be exempt from the recall bias. Moreover, the nature of the study as a cross-sectional study cannot elucidate the cause-effect relationship. Lastly, the study was based on self-reporting, which could have resulted in a social desirability bias.

## Conclusion

According to the current study, hypertension was at a higher level among clients receiving HAART for the planning and implementation of intervention measures. Factors significantly associated with hypertension in this study were: being on HAART for $\geq60$ months, high alcohol consumption, being on TDF/3TC/EFV, and TDF/3TC/NVP combinations. Emphasis should be placed on planning interventions to reduce hypertension in clients who have received TDF/3TC/EFV and TDF/3TC/NVP and clients who have received HAART for $\geq60$ months. Therefore, a concerted effort is required among health care providers to halt the habit of high alcohol use through counseling and education must be considered to reduce the occurrence of hypertension among PLWHIV. As this study is cross-sectional, it is hard to bring it to a close as clear effects of the current HAART regimen on hypertension. For that reason, these results could be supported by further study with analytic and experimental design.

## Supporting information

**S1 Dataset. The raw data supporting the findings of this article.**
(SAV)

**S1 Questionnaire. Data collection tool for the study.**
(DOCX)

## Acknowledgments

We are indebted to Hawassa University College of Medicine and Health Sciences School of Public Health for giving Ethical clearance to undertake the study. Our appreciation also goes to the managers and healthcare providers who worked in Nigist Elleni Mohamed Memorial Referral Hospital for their assistance and cooperation during the study. Finally, for their efforts, we want to thank our supervisors, data collectors, and study participants.

## Author Contributions

**Conceptualization:** Kaleegziabher Lukas, Endrias Markos, Akili Habte.

**Formal analysis:** Kaleegziabher Lukas, Fanuel Belayneh.

**Investigation:** Kaleegziabher Lukas, Akili Habte.

**Methodology:** Kaleegziabher Lukas, Endrias Markos, Fanuel Belayneh, Akili Habte.

**Project administration:** Kaleegziabher Lukas, Endrias Markos.

**Resources:** Kaleegziabher Lukas, Akili Habte.

**Software:** Akili Habte.

**Supervision:** Kaleegziabher Lukas, Endrias Markos, Fanuel Belayneh, Akili Habte.

**Writing – original draft:** Kaleegziabher Lukas, Akili Habte.

**Writing – review & editing:** Kaleegziabher Lukas, Endrias Markos, Fanuel Belayneh, Akili Habte.

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
