## [Decision Letter · Decision Letter 0]

11 Jan 2021

PONE-D-20-38405

The Magnitude of Hypertension and Associated Factors among Clients Receiving Highly Active Antiretroviral Treatment in Southern Ethiopia, 2020 (Hospital-Based Cross-Sectional Study)

PLOS ONE

Dear Dr. Habte,

Thank you for submitting your manuscript to PLOS ONE. After careful consideration, we feel that it has merit but does not fully meet PLOS ONE’s publication criteria as it currently stands. Therefore, we invite you to submit a revised version of the manuscript that addresses the points raised during the review process.

Please submit your revised manuscript by 6 Febraury If you will need more time than this to complete your revisions, please reply to this message or contact the journal office at plosone@plos.org. Please include the following items when submitting your revised manuscript:

We look forward to receiving your revised manuscript.

Kind regards,

Claudia Marotta

Academic Editor

PLOS ONE

Journal Requirements:

2. In statistical methods, please refer to any post-hoc corrections to correct for multiple comparisons during your statistical analyses. If these were not performed please justify the reasons. Please refer to our statistical reporting guidelines for assistance (https://journals.plos.org/plosone/s/submission-guidelines.#loc-statistical-reporting).

3.We suggest you thoroughly copyedit your manuscript for language usage, spelling, and grammar. If you do not know anyone who can help you do this, you may wish to consider employing a professional scientific editing service.  

Additional Editor Comments:

dear Authors follow reviewer suggestions to improve your paper

Reviewers' comments:

Reviewer's Responses to Questions

**Comments to the Author**

1. Is the manuscript technically sound, and do the data support the conclusions?

Reviewer #1: Partly

Reviewer #2: Yes

2. Has the statistical analysis been performed appropriately and rigorously? 

Reviewer #1: I Don't Know

Reviewer #2: No

3. Have the authors made all data underlying the findings in their manuscript fully available?

Reviewer #1: Yes

Reviewer #2: Yes

4. Is the manuscript presented in an intelligible fashion and written in standard English?

Reviewer #1: Yes

Reviewer #2: No

5. Review Comments to the Author

Reviewer #1: Major

1) the objectives of the study are not clear and should be stated in the initial study design section

2) the sample size calculation section is confusing. It should be clear what the primary objective is and the assumptions used to calculate a sample size to meet that primary objective.

3) The sampling strategy is also unclear. Were the clients listed randomly and then every third chosen? Were they stratified in any way? Or just every third chosen from the existing client register? This needs to be clarified in the manuscript.

4) Please address why no participants appear to be on second line PI-based regimens.

5) The frequency of hyperlipidemia is listed at 12 percent but it is not clear how this is defined or measured. Medical history or self report is unlikely to be accurate unless there is comprehensive screening for hyperlipidemia as standard of care in this setting.

6) It is not clear how the alternate measurement tool could account for the alcohol use finding. This should be addressed more specifically

7) The authors should discuss potential explanations for TDF based regimens being a risk factor and the implications of the TLD transition

8) I don't think use of contraceptives or receipt of corticosteroids are appropriate exclusion criteria. it would be important to know how common these exclusions were.

Minor

1) probably more appropriate to use the term "frequency" of hypertension rather than "magnitude"

2) Prefer "renal failure" to "kidney failure"

3) prefer "people living with HIV" to "HIV-positive people"

4) recommend consistently use ART instead of HAART

5) It would be helpful to describe the usual practice of blood pressure screening for HIV clinic visits at the hospital where participants were recruited and the usual standard of care for hypertension management in this setting.

6) It would be acceptable to just cite the methods for hypertension definition as opposed to detailing in the manuscript

7) Could consider using a separate table to list out the many definitions

8) describe what a recent viral load is considered to be in the paper (within the past year?)

9) What is the "reference" CD4? The nadir or baseline? Recommend clarification.

Reviewer #2: See attachment.

The paper needs language edit.

Issue related with sample size should be cleared

Current BMI should be in the model and see the AOD for other variables, the findings may change.

6. PLOS authors have the option to publish the peer review history of their article (what does this mean?). If published, this will include your full peer review and any attached files.

Reviewer #1: No

Reviewer #2: **Yes: **Saro Abdella Abrahim

---

## [Author Response · Author response to Decision Letter 0]

8 Feb 2021

The response to editor and reviewers' comments have been attached as a separate file as "Response to reviewers"

---

## [Editor Report · Decision Letter 1]

16 Aug 2021

PONE-D-20-38405R1

The Magnitude of Hypertension and Associated Factors among Clients on Highly Active Antiretroviral Treatment in Southern Ethiopia, 2020: A Hospital-Based Cross-sectional Study

PLOS ONE

Dear Aklilu Habte,

Thank you for submitting your manuscript to PLOS ONE. After careful consideration, we feel that it has merit but does not fully meet PLOS ONE’s publication criteria as it currently stands. Therefore, we invite you to submit a revised version of the manuscript that addresses the points raised during the review process.

We look forward to receiving your revised manuscript.

Kind regards,

Professor Kwasi Torpey, MD PhD MPH

Academic Editor

PLOS ONE

Journal Requirements:

Additional Editor Comments (if provided):

Thank you for the revision of the manuscripts to address the reviewers' comments. It has significantly improved however there are still a number of areas requiring attention.

1. There is a difference between the manuscript with tracked changes and the clean version. The tracked changes version captures most of the proposed changes. I suggest the authors to use the tracked changes version to generate the clean version paying attention to version control. Given the differences, the tracked changes version was used in this review

2. Abstract/Results: Correct typo hypertensiion to hypertension

3. Introduction page 3 last but one paragraph: Should be ART not ARRT

Page 3 last but one paragraph: Should be people living with HIV not people livening with HIV

4. Intro page 3 last but one paragraph : Sentence should read treatment is not risk free or free of side effects

and not risk free or side effects

5. Into last paragraph AIDS unrelated mortality should be non-AIDS related mortality

6. Materials and methods: need to be consistent on use of capitalization. For examples naming the departments ie Some are capitalized others are not

7. Population of the study line 172: remove comma and start capitalize w in women

8. Data quality management Line 236/7: . Revise sentence. Data collected was verified for completeness. PI and supervisor oversaw overall data collection. Remove every day from sentence

9. Conclusion: Revise opening statement . Seems out of place

10. Manuscript needs another round of copyediting. Lipodystrophy instead of lypodystrophy. Keep spelling consistent

11. Revise all references to meet journal requirements

---

## [Author Response · Author response to Decision Letter 1]

17 Aug 2021

The response to reviews have been attached as a "Response to Reviewers" in the

---

## [Editor Report · Decision Letter 2]

1 Oct 2021

The Magnitude of Hypertension and Associated Factors among Clients on Highly Active Antiretroviral Treatment in Southern Ethiopia, 2020: A Hospital-Based Cross-sectional Study

PONE-D-20-38405R2

Dear Aklilu Habte,

We’re pleased to inform you that your manuscript has been judged scientifically suitable for publication and will be formally accepted for publication once it meets all outstanding technical requirements.

Kind regards,

Professor Kwasi Torpey, MD PhD MPH

Academic Editor

PLOS ONE
---

## [Editor Report · Acceptance letter]

8 Oct 2021

PONE-D-20-38405R2 

The Magnitude of Hypertension and Associated Factors among Clients on Highly Active Antiretroviral Treatment in Southern Ethiopia, 2020: A Hospital-Based Cross-sectional Study 

Dear Dr. Habte:

I'm pleased to inform you that your manuscript has been deemed suitable for publication in PLOS ONE. Congratulations! Your manuscript is now with our production department. 

Kind regards, 

on behalf of

Professor Kwasi Torpey 

Academic Editor

PLOS ONE